# Mechanic Insight into the Distinct and Common Roles of Ovariectomy Versus Adrenalectomy on Adipose Tissue Remodeling in Female Mice

**DOI:** 10.3390/ijms24032308

**Published:** 2023-01-24

**Authors:** Weihao Chen, Fengyan Meng, Xianyin Zeng, Xiaohan Cao, Guixian Bu, Xiaogang Du, Guozhi Yu, Fanli Kong, Yunkun Li, Tian Gan, Xingfa Han

**Affiliations:** Isotope Research Lab, Biological Engineering and Application Biology Department, Sichuan Agricultural University, Ya’an 625014, China

**Keywords:** adipose tissue, remodeling, ovariectomy, adrenalectomy, transcriptomics, mice

## Abstract

Dysfunctions of the ovaries and adrenal glands are both evidenced to cause aberrant adipose tissue (AT) remodeling and resultant metabolic disorders, but their distinct and common roles are poorly understood. In this study, through biochemical, histological and RNA-seq analyses, we comprehensively explored the mechanisms underpinning subcutaneous (SAT) and visceral adipose tissue (VAT) remodeling, in response to ovariectomy (OVX) versus adrenalectomy (ADX) in female mice. OVX promoted adipocyte differentiation and fat accumulation in both SAT and VAT, by potentiating the *Pparg* signaling, while ADX universally prevented the cell proliferation and extracellular matrix organization in both SAT and VAT, likely by inactivating the *Nr3c1* signaling, thus causing lipoatrophy in females. ADX, but not OVX, exerted great effects on the intrinsic difference between SAT and VAT. Specifically, ADX reversed a large cluster of genes differentially expressed between SAT and VAT, by activating 12 key transcription factors, and thereby caused senescent cell accumulation, massive B cell infiltration and the development of selective inflammatory response in SAT. Commonly, both OVX and ADX enhance circadian rhythmicity in VAT, and impair cell proliferation, neurogenesis, tissue morphogenesis, as well as extracellular matrix organization in SAT, thus causing dysfunction of adipose tissues and concomitant metabolic disorders.

## 1. Introduction

Adipose tissue (AT) is a central metabolic organ in orchestrating whole-body energy homeostasis [1]. In addition to function as a passive fuel reservoir, AT is now recognized as an active endocrine organ and secretes various hormones and cytokines (termed as adipokines) to communicate with other organs, and thereby controls systemic energy balance and glucose homeostasis [2]. Generally, adipose tissue undergoes continuous dynamic changes in adipocyte and immune cell turnover, angiogenesis and extracellular matrix remodeling, so as to adapt the dynamic energy supply and physiological/pathophysiological alterations [3]. The manner in which the adipose tissue expands and remodels, directly impacts the risk of metabolic disorders and diseases [4].

According to the different anatomical depot location, white adipose tissue (WAT) is largely divided into subcutaneous adipose tissue (SAT) and visceral adipose tissue (VAT) within the body [5]. Dysfunction in different compartments of WAT has been associated with distinct metabolic disturbances. Generally, there is a consensus that VAT has a strong predisposition towards metabolic diseases, including metabolic syndrome, insulin resistance, type 2 diabetes, cardiovascular disease and steatosis [6,7,8], whereas SAT is more involved in metabolic improvements [9]. However, the underlying molecular mechanisms are poorly understood.

Alterations in adipose remodeling, including either adipose expansion or lipoatrophy, are tightly associated with the risk of metabolic disorders and diseases. Adipose tissue remodeling is regulated by various exogenous and endogenous factors, such as energy status and endocrine hormone fluctuations [2]. Among endocrine hormones, steroid hormones produced by both the gonads and adrenal glands play predominant roles in regulating adipose expansion and distribution. For example, loss of ovarian function, such as ovariectomy in rodent models or menopause in women causes increased fat deposition in VAT and other ectopic sites, which largely increases the risk of metabolic diseases [6,7,8]. For adrenal glands, strong evidence supporting its important roles in regulating AT remodeling is the adrenal insufficiency (Addison’s disease) and Cushing’s syndrome in humans [10]. Patients with Addison’s disease (cortisol deficiency) show no increase in adipose deposition, whereas patients that suffered from Cushing’s syndrome (hypercortisolism) have a centripetal adipose tissue distribution and an adverse adipokine profile [10]. Particularly and interestingly, patients with Cushing’s syndrome have increased lipolysis in SAT, but increased lipogenesis in VAT [11], highlighting depot-specific actions of adrenal glucocorticoids (GCs) on fat metabolism. Previous studies evidenced that adrenal gland-derived GCs play a central role in controlling nutrient substrate availability and systemic fuel partitioning, thereby essentially regulating the adipose expansion, distribution and whole-body energy homeostasis [12,13]. Very interestingly, recent studies on rodents reported that increased fat deposition induced by ovariectomy (OVX) could be completely prevented or reversed by adrenalectomy (ADX) [7], implicating the indispensable role of the adrenal gland-derived factors in OVX-induced fat deposition. However, how the adrenal glands coordinate with ovaries to co-regulate AT expansion and remodeling is largely unknown.

Loss-function of the ovaries or adrenal glands both causes AT dysfunction and metabolic disorders [6,7,8]. However, little is known about their distinct and common roles in regulating AT remodeling and resultant metabolic disorders. In this study, we conducted biochemical, histological and RNA-sequencing (RNA-seq) analyses to comparatively characterize the morphological and functional changes of WAT in both SAT and VAT depots from a sham control, ovariectomized and adrenalectomized female mice. Through independent and overlapping analyses of the RNA-seq data, as well as an integrative analysis of our data with other published omics data, we identified the distinct and common roles of OVX versus ADX in regulating the AT remodeling, as well as the resultant metabolic disorders.

## 2. Results

### 2.1. OVX and ADX Cause Opposite Changes in Endocrine Hormones and AT Expansion

To explore the effects of OVX versus ADX on the AT remodeling and metabolic disorders, female mice were ovariectomized or adrenalectomized at 6 weeks of age, and body weight was monitored weekly until the age of 30 weeks. Compared to the sham controls, the body weight of female mice was markedly increased (*p* < 0.001) following OVX, but remained unchangeable (*p* > 0.05) following ADX (Figure 1A). At decapitation, compared to the sham controls, both SAT and VAT mass was markedly increased (*p* < 0.001) in OVX mice but decreased (*p* < 0.05) in ADX mice (Figure 1B,C).

Sex hormones, e.g., 17β-estradiol [14,15] and follicle-stimulating hormone (FSH) [16,17,18], and adrenal gland-derived factors, e.g., glucocorticoids [11,13], are all evidenced to play pivotal roles in orchestrating AT remodeling as well as the resultant metabolic syndromes in females. Through an enzyme-linked immunosorbent assay (ELISA), we found that, compared to the sham controls, OVX decreased (*p* < 0.001) serum levels of 17β-estradiol, but increased (*p* < 0.001) FSH and corticosterone. In contrast, compared to OVX, ADX completely caused (*p* < 0.05) an opposite change in all of the three detected hormones in female mice (Figure 1D–F). In parallel, serum triglyceride (TG), free fat acid (FFA) and fasting blood glucose levels were elevated (*p* < 0.001) in mice following OVX, but remained comparable (*p* > 0.05) between ADX and the sham control mice (Figure 1D,E).

Consistently, OVX increased (*p* < 0.05) the adipocyte size in both VAT and SAT, but all of these parameters were lowered in female mice following ADX (Figure 1J–L). These results suggested that the opposite changes in serum 17β-estradiol, FSH and corticosterone levels might be important or even the primary reasons to cause the opposite change of the AT expansion and circulating lipid levels between OVX and ADX mice.

### 2.2. Transcriptomic Responses of AT to OVX versus ADX in Female Mice

To gain insight into the distinct remodeling response of AT to OVX versus ADX, RNA-seq was conducted on VAT and SAT samples from the sham control, OVX and ADX female mice. A total of 76.43 G clean bases from 12 samples were obtained with 92% of the bases scoring Q30. Hierarchical clustering comparing the patterns of gene expression indicated a striking separation between VAT and SAT across all treatment groups (Figure 2A), indicating there is a huge intrinsic difference between VAT and SAT, and this intrinsic difference could not be profoundly challenged by OVX- or ADX-induced physiological/pathophysiological changes. There was also a striking separation in the gene expression patterns between OVX and the sham control, ADX and the sham control, and between OVX and ADX, in both VAT and SAT (Figure 2A), revealing that both OVX and ADX caused a predominant and distinct AT remodeling in VAT and SAT.

Overlapping analysis of the differentially expressed genes (DEGs) (see Materials and Methods for criteria) among groups revealed distinct, common and core DEGs between VAT and SAT, in response to OVX (Figure 2B) and ADX (Figure 2C), respectively. A pairwise comparison between groups identified 324 and 2970 DEGs between OVX and the sham control, 1205 and 5232 DEGs between ADX and the sham control, and 654 and 4985 DEGs between OVX and ADX in VAT and SAT, respectively (Figure 2D and Appendix A). Quite interestingly, there were 9.2 and 4.3 times more DEGs in SAT than in VAT following OVX and ADX, respectively, implicating that SAT is more susceptible to OVX/ADX than VAT. Of these DEGs, a greater proportion of genes were downregulated in VAT (65.74%) and SAT (52.73%) in female mice following OVX, whereas a small proportion of DEGs was upregulated in both VAT (50.87%) and SAT (53.04%) in female mice following ADX (Figure 2D and Appendix A). This was somewhat opposite to the expansion direction of AT in mice following OVX or ADX.

The intrinsic difference between VAT and SAT was also profiled. Totally, there were 4171 DEGs between SAT and VAT in the sham control mice (Figure 2D and Appendix A). The number of DEGs between SAT and VAT was shortened to 1121 (3.7 times lower) following OVX treatment, but enlarged to 6586 (1.58 times more) following ADX treatment (Figure 2D and Appendix A), suggesting OVX and ADX exert distinct or even opposite actions on the intrinsic difference of AT between SAT and VAT.

### 2.3. Distinct and Common Functional Changes of VAT and SAT, in Response to OVX

In VAT, a pairwise comparison between groups identified 324 DEGs between OVX and the sham control, of which 111 (34.26%) DEGs were upregulated and 213 (65.74%) DEGs were downregulated in OVX mice (Figure 2D). A functional enrichment analysis using DAVID (2021 update) [19] showed that the upregulated DEGs in VAT were predominantly annotated into the biological process (BP) of regulation of the metabolic process, glucose metabolic process, oxidation-reduction process, phosphorylation, rhythmic process, response to oxygen levels, cell communication, fat cell differentiation, angiogenesis, etc. (Appendix A). While the downregulated DEGs were mainly enriched into the BP of the immune response, cytokine production, inflammatory response, cell adhesion, cell proliferation/differentiation regulation and intracellular signal transduction, as well as into the cellular component (CC) of the extracellular region and space, and into the molecular function (MF) of PI3K, a receptor and macromolecular binding complex (Appendix A). These results implicated that OVX promoted the AT expansion in VAT mainly through enhancing fat cell differentiation, angiogenesis and metabolic activities. However, the expansion of VAT, in turn, leads to concomitant immune inflammation, which might be one of important causes of AT dysfunction, as well as the resultant metabolic disorders in OVX mice.

In SAT, 2970 DEGs were identified between OVX and the sham control, of which 1404 (47.27%) DEGs were upregulated and 1566 (52.72%) DEGs were downregulated in OVX mice (Figure 2D). A functional enrichment analysis indicated that the upregulated DEGs were mainly enriched into the BP of the metabolism processes, including the oxidation-reduction process, carboxylic acid metabolic process, oxoacid metabolic process, organophosphate metabolic process, lipid metabolic process, nucleotide metabolic process, fatty acid metabolic process, coenzyme metabolic process, tricarboxylic acid process, oxidative phosphorylation, etc., into the CC of mitochondrion, into the MF of cofactor/coenzyme binding, electron carrier activity, NADH dehydrogenase activity, etc., and into the KEGG of metabolic pathways, oxidative phosphorylation, citrate (TCA) cycle, fatty acid degradation, HIF-1 signaling pathway, insulin resistance, etc. (Appendix A). The downregulated DEGs were mainly enriched into the BP of cell adhesion, cell migration, cell development/morphogenesis, tissue/tube morphogenesis, neurogenesis, cytoskeleton organization, extracellular organization, cell death, cytokine production, leukocyte differentiation, etc., into the CC of cell junction, extracellular matrix, synapse, etc., into the MF of structural molecular activity, cell adhesion molecular binding, extracellular matrix structural constituent, receptor binding, cytoskeletal protein binding, etc., and into the KEGG of ECM-receptor interaction, ribosome, focal adhesion, tight junction, PI3K-Akt/Wnt/estrogen signaling pathway, Cushing’s syndrome, etc. (Appendix A). Therefore, OVX appeared to profoundly enhance the metabolic activities of various nutrients/substrates, as well as TCA cycle activities in SAT and caused SAT aberrant remodeling by degenerating cell/tissue morphogenesis, neurogenesis, cytoskeleton organization and extracellular organization, thus resulting in metabolic disorders, such as insulin resistance and Cushing’s syndrome in mice.

To identify the key genes that mediate the OVX-induced AT aberrant remodeling and resultant metabolic disorders in OVX mice, we isolated the common up- and downregulated DEGs by OVX in both VAT and SAT. As a result, there were 68 common upregulated and 111 common downregulated DEGs in both VAT and SAT of mice following OVX (Appendix A). Of those 68 commons upregulated DEGs, a large subset of genes was associated with adipocyte differentiation (*Itga6*, *Lpin1*, *Lrrc8c*, *Mrap*, *Per2*, *Pparg*, *Zbtb16*), angiogenesis (*Apln*, *Arhgap24*, *Cspg4*, *Emcn*, *Pparg*), lipid metabolic process (*Acer2*, *Adora1*, *Fabp5*, *Gpd1*, *Hacd2*, *Lpin1*, *Nr1d2*, *Per2*, *Pparg*, *Vldlr*), oxidation-reduction process (*Gpd1*, *Idh3a*, *Mt-Co2*, *Mt-Co3*, *Per2*, *Pparg*, *Ppp1r3d*) and mitochondrion (*Abcc9*, *Cryzl2*, *Cyb5b*, *Gm10925*, *Gpd1*, *Idh3a*, *Ldhb*, *Lpin1*, *Mt-Co2*, *Mt-Co3*, *Mt-Rnr2*, *Mt-Tp*) (Appendix A). Thus, enhanced adipocyte differentiation and metabolic activities are the common cause of AT expansion in both VAT and SAT in mice following OVX. While, a functional enrichment analysis showed that the common downregulated DEGs were mainly annotated into the BP of the defense response, immune response, myeloid/leukocyte differentiation, cytokine production, regulation of cell death, inflammatory response, etc., into the CC of the extracellular region and space, into the MF of receptor binding, macromolecular complex and PI3K binding, and into the KEGG of antigen processing (Appendix A). Therefore, immune functions in both VAT and SAT depots were impaired and deteriorated during AT expansion following OVX, which may be an important or even the key cause of AT dysfunction and resultant metabolic disorders in females.

The expression of three genes (*Tnfrsf21*, *Postn* and *Plk2*) were upregulated in VAT but downregulated in SAT following OVX, compared to the sham controls (Appendix A). Augmented expression of the three genes was all reported to be associated with inflammation [20,21,22], implicating that OVX might cause more pronounced inflammation in VAT than SAT. In contrast, another three genes (i.e., *Slc1a3*, *B3galt2* and *Sorcs2*) were downregulated in VAT but upregulated in SAT following OVX, compared to the sham controls (Appendix A). Among them, *Slc1a3*, as a glutamate transporter, plays important roles in maintaining the electron transport chain and TCA cycle activity [23]; *B3galt2*, as a galactosyltransferase encoding gene, is associated with the extracellular matrix (ECM) remodeling [24]; while *Sorcs2* exerts a role in promoting angiogenesis [25], and protects cells from oxidative stress [26]. Therefore, SAT seemed to have a higher adaptability to OVX-induced AT expansion by promoting the TCA cycle activity, angiogenesis and extracellular matrix remolding.

### 2.4. Integrated Analysis Highlights the Central Role of Pparg in Mediating OVX-Induced AT Expansion

A protein-protein interaction (PPI) network analysis of all of the DEGs (i.e., all DEGs from Appendix A), using STRING APP of Cytoscape (V3.9.1), highlighted a central role of *Pparg* [with the highest number (133) of first connected nodes] (Figure 3A) in mediating the OVX-induced AT expansion and remodeling. It is well established that *Pparg* functions as a master transcription factor to regulate both adipogenesis and lipogenesis, thus playing a crucial role in orchestrating AT remodeling in mammals [27,28]. We further isolated the nodes (DEGs) centered on *Pparg* to construct a subnetwork (Figure 3B). A functional enrichment analysis was performed on these DEGs first connected to *Pparg* in SAT/VAT using DAVID (2021 update). Notably, the genes involved in adipogenesis and/or lipogenesis (e.g., *Lpin1*, *Klf15*, *Vldlr*, *Per2* and *Fabp5*) were significantly upregulated, whereas the genes involved in the insulin response (*Irs1*, *Egr2*, *Agt* and *Myc*) were all significantly downregulated by OVX, in both VAT and SAT, implicating OVX induced the AT expansion and associated insulin resistance through the potentiating *Pparg* signaling. The *Pparg* first connected DEGs selectively in SAT were all upregulated and mainly enriched into the lipid metabolism, fat cell differentiation, cell proliferation, transport and other metabolic process, also confirming the essential role of *Pparg* in mediating the OVX-induced fat expansion in SAT (Figure 3B). While, most of *Pparg* first connected DEGs selectively in VAT were downregulated and mainly enriched in angiogenesis and metabolic process (Figure 3B).

To further validate the downstream target genes of *Pparg* that mediate the OVX-induced AT expansion and remodeling, we conducted an overlapping analysis of our DEGs with the PPARG ChIP-seq data of a mouse 3T3-L1 cell line (SRX330315). As a result, 333 DEGs were identified to be the putative *Pparg* target genes (defined as genes with PPARG binding loci located within 3kb of its TSS) (Appendix A). Among those, nine and four genes were commonly upregulated and downregulated by OVX, respectively, in both VAT and SAT. Seven and six genes were selectively upregulated and downregulated by OVX, respectively, in VAT; 247 and 60 genes were selectively upregulated and downregulated by OVX, respectively, in SAT (Appendix A). Of those, *Mrap* and *Gpd1* were characterized as top upregulated DEGs during adipogenesis of 3T3-L1 cell in a multiple transcriptome analysis [29], and *Gpd1* is also important for lipogenesis by providing substrates for the synthesis of glycerol [30]. The expression of all of these *Pparg* target and key AT expansion-promoting genes were consistently upregulated by OVX, in both VAT and SAT, also highlighting the central role of *Pparg* in mediating the OVX-induced AT expansion. While, among the common downregulated *Pparg* target genes, both *Irs1* and *Myc* are involved in insulin signaling (Figure 3D), revealing the *Pparg*-mediated insulin resistance in females during ovarian insufficiency. Additionally, a functional enrichment analysis showed that the 247 *Pparg* target DEGs selectively upregulated in SAT, were predominantly annotated into the lipid metabolism, including lipid transport/storage, fatty acid biosynthesis, TCA cycle, etc. (Appendix A), again reinforcing the central role of *Pparg* in mediating the OVX-induced AT expansion and remodeling in SAT.

### 2.5. Distinct and Common Functional Changes of VAT and SAT, in Response to ADX

In VAT, the pairwise comparison between groups identified 6586 DEGs between ADX and the sham control mice, of which 2950 (44.79%) were upregulated and 3636 (55.21%) were downregulated in ADX mice, respectively (Figure 2D). The functional enrichment analysis indicated that the upregulated DEGs were mainly annotated into the BP of the carboxylic acid/oxoacid/nucleotide/lipid/ATP/fatty acid/coenzyme metabolic process, oxidative phosphorylation, etc., into the CC of the mitochondrion, mitochondrial protein complex, oxidoreductase complex, respiratory chain, peroxisome, etc., into the MF of the coenzyme/cofactor/ion binding, electron carrier activity, etc., and into the KEGG of the metabolic pathways, carbon/fatty acid/pyruvate metabolism, oxidative phosphorylation, citrate cycle, etc. (Appendix A). The downregulated DEGs were mainly annotated into the BP of the defense response, innate immune response, cell adhesion, cell migration, extracellular structure organization, cytokine production, vasculature development, cell proliferation, neurogenesis, etc., into the CC of the extracellular region/matrix, into the MF of the extracellular matrix structural constituent, protein complex binding, anion/integrin binding, etc., and into the KEGG of the ECM-receptor interaction, focal adhesion, phagosome and PI3K-Akt signaling pathway (Appendix A). These results suggest that ADX predominantly enhanced various nutrient metabolism and mitochondrion functions, but impaired the immune function, vasculogenesis and neurogenesis in VAT.

In SAT, 5232 DEGs were identified between ADX and the sham control mice, of which 2775 (53.04%) were upregulated and 2457 (46.96%) were downregulated in ADX mice (Figure 2D). The functional enrichment analysis indicated that the upregulated DEGs were predominantly annotated into the BP of the immune response, leukocyte activation, adaptive immune response, regulation of lymphocyte activation/immune response, B/T cell activation, phagocytosis, cytokine production, complement activation, inflammatory response, cell cycle, cell adhesion, DNA replication, interleukin-6/10/12 production, etc., into the CC of the immunoglobulin complex, extracellular space, chromosome, replication fork, mitotic spindle, etc., into the MF of the antigen/immunoglobulin receptor/enzyme/MHC protein complex/kinase/anion binding, cytokine receptor activity, etc., into the KEGG of the primary immunodeficiency, B cell receptor signaling pathway, cytokine-cytokine receptor interaction, NF-kappa B signaling pathway, T cell receptor signaling pathway, Th1/Th2/Th17 cell differentiation, DNA replication, nature killer cell mediated cytotoxicity, cell cycle, etc. (Appendix A). While, the downregulated DEGs were predominantly annotated into the BP of the cell adhesion, cell migration, anatomical structure morphogenesis, cell development, neurogenesis, blood vessel development, into the CC of the extracellular matrix and cell junction, into the MF of the cell adhesion molecular/cytoskeletal protein binding, and into the KEGG of the ECM-receptor interaction, focal adhesion, Pl3K-Akt singling pathway, Wnt signaling pathway, etc. (Appendix A). These results suggest that ADX predominantly promoted the immune cell infiltrations into SAT and thus resulted in tissue inflammation. ADX caused SAT lipoatrophy mainly by suppressing the cell development/morphogenesis, neurogenesis and blood vessel development. Especially, the Pl3K-Akt and Wnt signaling pathways might play important roles in mediating the ADX-induced lipoatrophy in SAT.

To gain insight into the common mechanisms by which ADX essentially regulates the AT aberrant remodeling in females, the common up- and downregulated DEGs by ADX in both VAT and SAT were isolated. In total, there were 101 common upregulated and 270 common downregulated DEGs in both VAT and SAT of mice following ADX (Appendix A). The functional enrichment analysis showed that the common upregulated DEGs were predominantly annotated into the BP of the B cell activation, B cell mediated immunity, humoral immune response, phagocytosis, lipid metabolic process, acyl-CoA metabolic process, leukocyte activation, etc., into the CC of the immunoglobulin complex, extracellular space, etc., into the MF of the immunoglobulin receptor/antigen binding, and into the KEGG of the metabolic pathways, pyruvate metabolism and fatty acid metabolism (Appendix A). While the common downregulated DEGs were mainly annotated into the BP of the extracellular matrix organization, cell adhesion, cell migration, vasculature development, organ/tissue morphogenesis, neurogenesis, etc., into the CC of the extracellular matrix/region and cell junction, into the MF of the extracellular matrix structural constituent, structural molecular activity, protein complex binding, and into the KEGG of the ECM-receptor interaction, focal adhesion, protein digestion and absorption, as well as the PI3K-Akt signaling pathway (Appendix A). Collectively, these results suggested that ADX caused lipoatrophy in both SAT and VAT, predominantly by suppressing the extracellular organization, cell adhesion, vascular devolvement, organ/tissue morphogenesis and neurogenesis.

Furthermore, ADX caused a GC deficiency, and GCs play a great role in modulating the AT remodeling through binding and activating its cognate receptor *Nr3c1*, a ligand dependent nuclear transcription factor [31]. We then performed a PPI network analysis using all of the ADX-induced DEGs in VAT and SAT (i.e., all of the DEGs from Appendix A), and the subnetwork constructed by the DEGs first connected to *Nr3c1* were extracted (Figure 4). As a result, 127 DEGs were centered on *Nr3c1*, including 18 common DEGs in both VAT and SAT, 18 VAT-specific DEGs and 92 SAT-specific DEGs. The functional enrichment analysis indicated a GC deficiency universally suppressed the cell proliferation and promoted protein targeting and modification in both VAT and SAT, but conversely affected the immune response between VAT and SAT (Figure 4). In VAT, the GC deficiency promoted the metabolic process but suppressed the response to stimulus; while in SAT, the GC deficiency suppressed the transcription, cell cycle, anatomical structure formation and cytoskeleton, but promoted the metabolic regulation (Figure 4). Through an overlapping analysis of these DEGs centered on *Nr3c1* with previously published NR3C1 ChIP-seq data in murine 3T3-L1 cells [32], *Fkbp5*, *Per2*, *Adrb2*, *Angptl4*, *Mc2r*, *Dusp1*, *Tle1* and *Fosl2* were identified to be *Nr3c1* target genes, revealing that these genes may play essential roles in mediating ADX-induced AT aberrant remodeling, as well as the resultant metabolic disorders.

### 2.6. ADX, but Not OVX, Exerts Great Effects on the Intrinsic Difference between SAT and VAT

Great intrinsic differences exist between VAT and SAT, either for their rates of hypertrophy and hyperplasia during growth and remodeling, or for their dysfunction or inflammation in response to obesity [33,34,35,36]. The pairwise comparison revealed 4171 DEGs between VAT and SAT in the sham control female mice, of which 1909 (45.77%) were upregulated and 2262 (54.23%) were downregulated in VAT compared to SAT (Figure 2B). The functional enrichment analysis indicated that the VAT-enriched genes (i.e., the upregulated DEGs in VAT) were mainly annotated into the BP of the lipid metabolic process, angiogenesis and cell death, into the CC of the mitochondrion, peroxisome and endoplasmic reticulum, into the MF of the cofactor/coenzyme/small molecule/ion binding, and into the KEGG pathway of the fatty acid metabolism, PPAR and insulin signaling pathways (Figure 5A). In contrast, the SAT-enriched genes (i.e., the upregulated DEGs in SAT) were mainly enriched in the BP of the epithelium development/differentiation, tube development, neurogenesis, etc., and into the KEGG of the Wnt signaling pathway and PI3K-Akt signaling pathway (Figure 5B). These results revealed that VAT is metabolically more active than SAT. Indeed, all of the key genes involved in the lipid metabolism (Figure 5C), and most of the genes encoding adipokines (Figure 5D), were more highly expressed in VAT than in SAT. Notably, in parallel with the Wnt signaling pathway enriched in SAT, we found that multiple *Wnt* genes, including *Wnt2*, *Wnt4*, *Wnt5b* and *Wnt7b* were more highly expressed in SAT than VAT (Figure 5E). It is well known that the Wnt signaling inhibits the recruitment of adipose precursor cells and their commitment to the adipogenic lineage [37,38], Particularly, *Wnt2* was recently reported to restrain the commitment of adipocytes specifically in SAT [39]. Thus, the activation of the Wnt signaling pathway in SAT might be a determinant factor to limit its adipogenic capacity and thereby lead to its lower expansion potential than that of VAT. Interestingly, the highly activated Wnt signaling seems to be a major intrinsic trait of SAT as it could not be easily challenged upon OVX or ADX stimuli (Figure 5F). However, OVX seemed to largely change the intrinsic difference in the lipid metabolism between SAT and VAT (Figure 5C).

To further explore the impact of OVX versus ADX on the intrinsic difference between VAT and SAT, we performed an overlapping analysis of the DEGs comparing VAT to SAT between the OVX mice and sham controls, and between the ADX and sham controls. As a result, OVX exerted minimal effects on the intrinsic difference between VAT and SAT, as evidenced by almost no conversely expressed common DEGs (except for three genes), comparing VAT to SAT between OVX and the sham controls (Figure 6A). In stark contrast, there was a great number of conversely expressed common DEGs, comparing VAT to SAT between ADX mice and the sham controls (Figure 6B), suggesting ADX exerted a profound impact on the intrinsic difference between VAT and SAT.

Interestingly, 78 genes, which were originally lower expressed in VAT than SAT in sham controls, were reversed upon ADX (marked by orange square in Figure 6B). Functional enrichment analysis indicated that these genes were mainly enriched into neurogenesis, cell development, morphogenesis and extracellular matrix organization (Figure 6C). While another 93 genes, which were originally less expressed in SAT than VAT in the sham controls, were reversed by ADX (marked by the green square in Figure 6B), and these genes were predominantly annotated into inflammation-associated terms (Figure 6D). Additionally, using FPKM > 1 as criteria for the gene expression [40], we identified 1054 out of 6586 DEGs, comparing VAT to SAT in ADX females, which were only selectively expressed in SAT (Appendix A). The functional enrichment analysis showed that these SAT selectively expressed DEGs were also predominantly involved in the inflammatory reaction mediated by a variety type of immune cells (Appendix A), revealing ADX predominantly and selectively caused inflammation in SAT. Indeed, a histology analysis evidenced that a large number of crown-like structures (CLSs), that are the histologic hallmarks of the proinflammatory process [41], were selectively detected in SAT rather than VAT, in ADX mice (Figure 6E).

We then attempted to identify the key genes that mediate the ADX-induced inflammation selectively in SAT. A transcription factor enrichment assay (TFEA), performed by ChIP-X Enrichment Analysis Version 3 (ChEA3) [42] indicated that these 93 ADX-reversed and inflammation-associated genes were mainly transcriptionally regulated by 12 transcription factors (TFs), i.e., *Sp140*, *Irf8*, *Scml4*, *Tbx21*, *Batf*, *Foxp3*, *Akna*, *Ikzf3*, *Pou2f2*, *Nfatc2*, *Gfi1* and *Sp110* (Figure 7A). These 12 TFs were all characterized as upregulated DEGs by ADX in SAT (Figure 7A), but not affected by ADX in VAT (except for *Sp110*); and they were also not differentially expressed between VAT and SAT in the sham controls (except for *Scml4*). Consistently, the literature research also confirmed that most of these TFs, such as *Irf8* [43], *Tbx21* [44], *Scml4*, *Sp140* and *Ikzf3* [45], were reported to be involved in proinflammatory responses in adipose tissues or other tissues. These results strongly suggest that ADX selectively caused inflammation in SAT. The cell-type-specific resolution gene expression atlas from *Tabula Muris* [46], based on single-cell RNA sequencing, further revealed that these TF coding genes were mainly expressed in immune cells, such as the B cell, T cell, NK cell and myeloid cell of adipose tissues (Figure 7C–O) and formed a tight TF co-regulatory network (Figure 7B).Therefore, ADX seemed to selectively promote the infiltration of massive proinflammatory cells, especially B cells, and resultantly caused inflammation in SAT by mainly upregulating these 12 TFs.

To estimate the type and percentage of the immune cell compositions in the AT of mice following ADX/OVX, a machine learning tool, CIBERSORTx [47], was employed to deconvolute our bulk RNA sequencing data. As a result, a large number of immune cell types were identified (Figure 8A). Of them, predominantly, the maximum percentage of the increased immune cell type in SAT, in mice following ADX, were B cells (Figure 8A,B), revealing a massive B cell infiltration that might be the key cause to contribute inflammation in SAT following ADX.

Moreover, we also noticed that the cellular senescence marker genes (including *Trp53*, *Cdkn2a* and *Cdkn1b*) (Figure 9A) [48], as well as a set of senescence related genes (Figure 9B) [49], were significantly and selectively upregulated in SAT upon ADX. To further validate the ADX-induced cellular senescence in SAT, we performed senescence-associated-β-galactosidase (SA-β-gal) staining to detect the senescent cell accumulation in both SAT and VAT, following OVX versus ADX. As expected, the level of SA-β-gal-positive cell staining was predominantly and selectively increased in SAT, rather than VAT following ADX (Figure 9C). However, OVX had also no obvious effects on the cell senescence in both SAT and VAT, compared to the sham controls (Figure 9C). Cellular senescence is a phenomenon characterized by the cessation of cell division. Senescent cell accumulation has been demonstrated to induce inflammation [49,50]. Therefore, senescent cell accumulation may be a key signal to trigger a massive inflammatory immune cell (especially B cell) infiltration and thus the development of inflammation in SAT following ADX.

### 2.7. Common Response of VAT/SAT to OVX and ADX

An overlapping analysis indicated that there existed a great number of DEGs in VAT/SAT commonly regulated by both OVX and ADX. In VAT, 46 genes were commonly and consistently upregulated, and 103 genes were commonly and consistently downregulated by both OVX and ADX, and just a few genes were commonly but reversely regulated by OVX and ADX (Appendix A). A functional enrichment analysis indicated that the commonly and consistently upregulated genes were predominantly enriched in the rhythmic process and small molecular biosynthetic process (Appendix A); while the commonly and consistently downregulated genes were mainly enriched in the immune response associated terms (Appendix A). These results highlighted that OVX and ADX could both orchestrate the VAT remodeling through enhancing its circadian rhythmicity and small molecular biosynthesis, but meanwhile impair its immune functions.

In SAT, there were 124 commonly and consistently upregulated genes, and 854 genes were commonly and consistently downregulated by both OVX and ADX (Appendix A). The functional enrichment analysis indicated that these commonly and consistently upregulated genes were predominantly annotated into metabolic processes including oxoacid, coenzyme, fatty acid, carboxylic acid, pyruvate and acyl-CoA metabolism (Appendix A); while the commonly and consistently downregulated genes were mainly enriched into the BP of the cell development/proliferation/adhesion, neurogenesis, organ/tissue/tube morphogenesis, extracellular matrix organization, etc., and into the KEGG of the PI3K-Akt signaling pathway, focal adhesion and ECM-receptor interaction (Appendix A). These results suggest that both OVX and ADX could commonly activate the metabolic activities of various substrates, but suppressed the cell development/differentiation, neurogenesis and tissue morphogenesis in SAT. In contrast to VAT, OVX and ADX also exerted an opposite action on the expression of a large number of common genes in SAT, there were 135 genes which were upregulated by OVX but downregulated by ADX (Figure 7C). The functional enrichment analysis showed that these genes were mainly enriched in the fatty acid metabolic process, fat cell differentiation and angiogenesis (Appendix A), consistent with our above demonstrated fact that OVX promotes but ADX inhibits fat accumulation in SAT. Furthermore, 132 genes were downregulated by OVX but upregulated by ADX, and this portion of genes was mainly annotated into the immune response, programmed cell death, leukocyte activation, cytokine production, cell chemotaxis, etc. (Appendix A), again reinforcing that ADX predominantly causes immune inflammation in SAT.

### 2.8. Distinct Response of VAT/SAT to OVX and ADX

To characterize the distinct roles of OVX versus ADX in regulating the AT remodeling and functioning, we isolated the distinct DEGs induced by OVX/ADX, in both VAT and SAT. In VAT, 168 genes were regulated (62 upregulated and 106 downregulated) by OVX but not by ADX (Appendix A), while 1049 genes that were regulated (561 upregulated and 488 downregulated) by ADX, but not by OVX (Appendix A), were isolated. Comparatively, the OVX distinctly upregulated DEGs in VAT were mainly involved in angiogenesis and phosphorylation (Appendix A); while the OVX distinctly downregulated DEGs in VAT were mainly enriched into the BP of the response to steroid hormone, ion transport, cell death, etc. (Appendix A). While, the ADX distinctly upregulated DEGs in VAT were mainly enriched into the BP of the carboxylic acid metabolic process, nucleotide metabolic process, cofactor metabolic process, lipid metabolic process and oxidative phosphorylation (Appendix A); and the ADX distinctly downregulated DEGs in VAT were mainly enriched in the BP of the immune response, extracellular matrix organization, angiogenesis, neurogenesis, tissue morphogenesis, etc. (Appendix A).

In SAT, 1725 genes were regulated (1145 upregulated and 580 downregulated) by OVX, but not by ADX (Appendix A), while 3788 genes were regulated (2320 upregulated and 1468 downregulated) by ADX, but not by OVX (Appendix A). The functional enrichment analysis indicated that the OVX distinctly upregulated DEGs in SAT were mainly enriched into the BP of the oxidative phosphorylation, lipid metabolic process, nucleotide biosynthetic process, fat cell differentiation, ion transport, etc. (Appendix A); and, the OVX distinctly downregulated DEGs in SAT were mainly enriched into the BP of the translation, gene expression, regulation of signal transduction, etc. (Appendix A). While the ADX distinctly upregulated DEGs in SAT were mainly enriched in the immune and inflammatory responses (Appendix A); and the ADX distinctly downregulated DEGs in SAT were mainly enriched into the BP of the anatomical structure/organ/tissue morphogenesis, angiogenesis, neurogenesis, etc. (Appendix A).

### 2.9. Integrative Network Analysis Reveals the Fat-Depot DIVERGENCE between OVX and ADX in Regulating the AT Remodeling

Our above data indicated that OVX and ADX exerted their actions on regulating the AT remodeling and resultant metabolic disorders, likely by principally potentiating the *Pparg* signaling and inactivating the *Nr3c1* signaling, respectively. However, the interaction or crosstalk between the two different signalings in regulating the AT remodeling is unknown. Through the integrative network analysis of our RNA-seq data, we found there existed a coordinative action between potentiating the *Pparg* signaling and inactivating the *Nr3c1* signaling in regulating the AT remodeling (Figure 10A). Particularly, in both VAT and SAT, OVX and ADX could commonly downregulate *Irs1* to impair the insulin signal transduction by potentiating the *Pparg* signaling and inactivating the *Nr3c1* signaling, respectively (Figure 10A). Then, in SAT, OVX, promoted but ADX suppressed expression of lipid metabolism-associated genes (i.e., *Lpl*, *Fabp4*, *Adipoq* and *Cav1*) by potentiating the *Pparg* signaling and inactivating the *Nr3c1* signaling, respectively, ultimately causing the opposite AT expansion and remodeling (Figure 10A). Very interestingly, in most situations, OVX and ADX converged on the same genes to drive their expressions in the same direction (i.e., either common-upregulation or common-downregulation) but at different AT depots through potentiating the Pparg and inactivating the Nrc3c1 signalings, respectively (Figure 10A), highlighting a depot divergence between OVX and ADX in regulating the lipid metabolism-associated gene expression and AT remodeling.

The distinct and common roles of OVX versus ADX in regulating the SAT/VAT remodeling were further summarized (Figure 10B). In summary, OVX or ovarian insufficiency, promoted the fat accumulation and AT expansion, and caused concomitant hyperglycemia by mainly potentiating the *Pparg* signaling in both VAT and SAT; in contrast, ADX or adrenal insufficiency, universally suppressed the cell proliferation and thus caused lipoatrophy, likely by activating the *Nr3c1* signaling in both VAT and SAT. Strikingly, ADX selectively caused the cellular senescence and massive B cell infiltration into SAT, consequently resulting in an immune inflammation in SAT. Distinctly, in SAT, OVX profoundly suppressed the gene transcription and translation, while ADX predominantly caused inflammation; and in VAT, OVX selectively promoted angiogenesis, while ADX disrupted the extracellular matrix organization and immune response, but augmented the metabolic-promoting signaling pathways. Commonly, in SAT, both OVX and ADX augmented the metabolic-promoting pathways, but impaired the cell proliferation, neurogenesis, tissue morphogenesis and extracellular matrix organization, thus causing aberrant SAT remodeling; and in VAT, both OVX and ADX enhanced the intrinsic circadian rhythmicity, thus impacting or disrupting the VAT function.

## 3. Discussion

Both the ovaries and adrenal glands play pivotal roles in AT remodeling and whole-body energy homeostasis, and dysfunctions of the two organs are both linked to aberrant AT remodeling and metabolic diseases. In parallel with the opposing metabolic phenotypes between menopausal women [16,51] and Addison’s disease patients [52,53], the loss-function of ovaries caused an expansion of both VAT and SAT with concurrent hyperlipidemia and hyperglycemia, whereas the loss-function of the adrenal glands caused lipoatrophy in both VAT and SAT with concomitant senescent cell accumulation and predominant inflammation, selectively in SAT. To the best of our knowledge, this was the first study to comprehensively characterize the distinct and common roles of ovarian insufficiency versus adrenal insufficiency in regulating the AT remodeling and resultant metabolic disorders.

Menopause in women or ovariectomy in rodents both cause AT expansion accompanied with hyperlipidemia and hyperglycemia [54,55], but the underlying molecular mechanisms are not well understood. By leveraging the RNA-seq, we identified distinct DEGs in VAT/SAT and common DEGs in both VAT and SAT in mice regulated by OVX. Functional enrichment analyses reveal that the activation of the fat cell differentiation, as well as the lipid biosynthesis and storage-promoting pathways are the common causes of AT deposition in both VAT and SAT in females, upon the loss of ovarian function. Through an integrative network analysis, we identified that *Pparg* played a central role in mediating the OVX-induced AT expansion, as well as the resultant hyperglycemia. Through bioinformatic analyses, we further revealed that OVX likely promotes the fat deposition through upregulating *Klf15*, *Per2*, *Lpin1*, *Vldlr* and *Fabp5*, and reduces the response to insulin through downregulating *Irs1* in both VAT and SAT, by potentiating the *Pparg* signaling. To identify the direct target genes that mediate *Pparg* to promote the OVX-induced AT expansion and resultant hyperglycemia, we performed overlapping analyses of our OVX-induced DEGs with other PPARG ChIP data from 3T3-L1 cells (SRX330315). As expected, in both VAT and SAT, a cluster of DEGs were the Pparg putative target genes. Particularly, among those, *Mrap* and *Gpd1* were both identified as top upregulated DEGs during the adipogenesis of 3T3-L1 cells [29], while *Irs1,* an encoding insulin receptor substrate, plays a central role in the insulin-signaling pathways [56] and *Myc* was reported to prevent adipogenesis [57]. Therefore, *Mrap*, *Gpd1*, *Irs1* and *Myc* appear to play key roles in transmitting the *Pparg* signal to promote the AT expansion and concomitant insulin resistance in females, upon the loss of ovarian function.

Unexpectedly, compared to VAT, there was a greater number of unique OVX-induced *Pparg* putative target DEGs in SAT. The functional analyses indicated that these SAT-selective *Pparg* putative target DEGs were predominantly enriched in the fatty acid metabolism, lipid transport and storage, suggesting OVX exerted much greater effect on SAT than VAT, by potentiating the *Pparg* signaling. It is well established that the increased fat deposition in response to energy surplus or loss of ovarian function first takes place in SAT [6]. When storage capacity is exceeded, fat is stored outside the SAT in the so-called ectopic fat depots which are important contributors to the obesity-associated insulin resistance and other metabolic disorders [7,8]. Therefore, it appears that potentiating the *Pparg* signal in SAT is the primary cause to drive the fat deposition in AT, as well as other ectopic depots, which subsequently lead to metabolic disorders, including dyslipidemia and hyperglycemia in females, upon the loss of ovarian function.

Contrary to the OVX-induced AT expansion, the loss of adrenal function caused lipoatrophy in both VAT and SAT in female mice, as evidenced by the decreased adipocyte cell size and fat pad weight. These results suggest that ovaries and adrenal glands basically exert opposite functional roles in orchestrating the AT expansion and remodeling. A transcriptomic analysis revealed that the common downregulated DEGs by ADX in both VAT and SAT, were predominantly enriched in the cell proliferation, organ/tissue morphogenesis, angiogenesis and neurogenesis. Obviously, all of these biological processes are necessary for tissue morphogenesis and functional maintenance. Therefore, the AT structural and morphological degeneration itself, rather than reduced adipogenesis/lipogenesis, appeared to be the primary cause to drive lipoatrophy in both SAT and VAT in females by ADX. Indeed, key genes involved in adipogenesis and/or lipogenesis such as *Mlxipl*, *Pparg*, *Acaca*, *Fasn*, *Elovl6*, *Scd1* and *Dgat2,* were either upregulated or unchanged rather than downregulated by ADX in both VAT and SAT, also ruling out the possibility that AT atrophy in ADX mice was a consequence of decreased adipogenesis or lipogenesis.

Glucocorticoids (GCs) secreted from the adrenal glands are one of the most important factors in regulating the AT remodeling, causing lipoatrophy when deficient, but promoting deposition with chronic exposure [58]. The GCs’ physiological actions are realized through binding to its cognate receptor, GR (NR3C1) which is a ligand dependent nuclear transcription factor [31]. Through an integrated PPI network analysis, we found that inactivating the *Nr3c1* signaling by ADX profoundly blocked the cell proliferation in both VAT and SAT, implicating that a cell proliferation blockage should play an essential role in mediating the ADX-induced AT atrophy. Particularly, inactivating the *Nr3c1* signaling by ADX, selectively disrupted the cell cycle, gene expression, anatomical structure development, cytoskeleton and metabolic regulation in SAT. Thus, similar to ovarian insufficiency, adrenal insufficiency also exerted more intensive effects or more detrimental impact on SAT than VAT by inactivating the *Nr3c1* signaling.

Previous studies have clearly evidenced that a great intrinsic or inherent difference exists between SAT and VAT, which is crucial for their distinct capacities for expansion, as well as susceptibilities to metabolic dysfunction [33,59]. In good agreement, in the present study, there existed a huge number of DEGs between VAT and SAT in female mice, irrespective of experimental treatments. Interestingly, many key genes involved in the lipid metabolism, including adipogenesis (*Cebpa*, *Pparg*), lipogenesis (*Fasn*, *Acaca*, *Acacb*, *Plin5*, *Lpl*, *Scd1*, *Scd3*, *Lpin1*) and fatty acid uptake/transport (*Cd36*, *Fabp4*) were all more highly expressed in VAT than in SAT, suggesting VAT is metabolically more active than SAT. Indeed, in the sham control females, the functional analyses also indicated that VAT-enriched genes were mainly enriched in the lipid metabolic processes and pathways, whereas SAT-enriched genes were mainly enriched in the non-metabolic pathways, such as cell development/adhesion, neurogenesis and cytoskeleton organization. Moreover, we found that multiple Wnt genes were more highly expressed in SAT than VAT. The higher expression pattern of the Wnt signaling genes in SAT than VAT still remained unvaried in both OVX and ADX situations, suggesting that neither an AT expansion or atrophy could change this inherent difference between SAT and VAT depots. Wnt signaling is well known to play essential roles in inhibiting the recruitment of adipose precursor cells, and they are commitment in the adipogenic lineage [37,38]. Thus, the higher expression of the Wnt signaling genes in SAT than in VAT should be a determinant factor to limit its adipogenic capacity, therefore inherently rendering its lower expansion potential than that of VAT.

Noteworthy, from reversing the gene expression profile standpoint, we found ADX, but not OVX, exerted a profound effect on the intrinsic difference between VAT and SAT. It seems that reversing the expression pattern of these intrinsic DEGs between VAT and SAT is detrimental to the AT physiology and function. Specifically, a large proportion of genes, which were initially more highly expressed in SAT, were reversed to be more highly expressed in VAT, in mice following ADX. The functional analysis indicated that these genes were predominantly enriched into tissue morphogenesis associated biological processes, such as neurogenesis, extracellular matrix organization and tissue morphogenesis, highlighting that adrenal-derived factors tend to favor SAT but inhibit VAT morphogenesis and functioning.

Very interestingly, another large proportion of genes, which were initially more highly expressed in VAT, were reversed to be more highly expressed in SAT, in mice following ADX. The functional analysis indicates that these genes were predominantly associated with inflammation, revealing ADX tended to induce inflammation in SAT. Indeed, a large number of crown-like structures (CLSs), the histologic hallmarks of the proinflammatory process [41], were selectively observed in SAT but not VAT, in ADX mice. Meanwhile, we also noted that the top upregulated DEGs between VAT and SAT by the ADX (i.e., *Cd19* and *Ms4a1*) encoding surface marker of the B cell lineage, thereby suggesting a possible role for B cells in SAT inflammation.

To gain insight into the mechanisms by which ADX selectively induced inflammation in SAT, a transcription factor enrichment assay (TFEA) was conducted for these ADX-reversed and inflammation-associated genes. As a result, we found that these genes were mainly transcriptionally regulated by 12 transcription factors (TFs). scRNA-seq data from *Tabula Muris* further validated that these TFs were highly expressed in immune cells, especially in B cells of AT. In parallel, a CIBERSORTx analysis also revealed that the B cells were strikingly and selectively increased in SAT of mice following ADX. T cells [60] and macrophages [61] in adipose tissue have been well documented for mediating AT inflammation. However, a recent study also highlighted the pivotal role of adipose B cells in AT inflammation and metabolic disease [62,63,64]. Therefore, herein we propose that the increased B cell infiltration might be the primary cause of inflammation in SAT, upon the loss of adrenal function.

Surprisingly, we found that cellular senescence marker genes were profoundly more highly expressed in SAT than VAT in female mice with ADX, implicating that ADX selectively promoted cell senescence in SAT. To further validate this speculation, we performed senescence-associated-β-galactosidase (SA-β-gal) staining to detect the senescent cell accumulation in both SAT and VAT, following OVX/ADX. As expected, the level of SA-β-gal-positive cell staining was predominantly and selectively increased in SAT, in mice following ADX. OVX had almost no obvious effects on cell senescence in both SAT and VAT, compared to the sham controls, as well. Cellular senescence is a phenomenon characterized by the cessation of cell division, and the senescent cell accumulation has been demonstrated to induce AT inflammation and adipocyte cell death by secreting proinflammatory cytokines [65,66]. The elimination of senescent cells in AT using senolytic agents, could effectively reduce inflammation in AT [49]. Moreover, aging-associated B cells, together with pro-inflammatory IgG and the cytokines they produced, have been reported to induce inflammation and dysmetabolism in adipose tissue [67]. Therefore, senescent adipocyte accumulation may be a key signal to trigger a massive B cell infiltration and thus cause a strong inflammatory response in SAT, following ADX. However, the specific mechanisms by which ADX upregulated those 12 core TFs and promoted the B cell infiltration selectively in SAT, are still unknown and require further investigation.

Although OVX and ADX exerted an opposite action on AT deposition and remodeling, they still both regulated a large number of common genes in the same direction in both VAT and SAT. The functional analysis indicated that both OVX and ADX strikingly enhanced the intrinsic circadian rhythmicity in VAT but caused structural and morphological degeneration in SAT by suppressing cell development/proliferation, tissue morphogenesis, neurogenesis, vasculature development and extracellular matrix development. Dysfunction of the circadian rhythmicity in AT has been suggested to aberrate the AT function and metabolism [68]. Therefore, it seems that both OVX and ADX commonly cause aberrant AT remodeling in VAT by disrupting its circadian rhythmicity, and in SAT by impairing its structural and morphological bases.

In summary, OVX and ADX exerted opposite actions on AT deposition, likely by potentiating the *Pparg* signaling and inactivating the *Nr3c1* signaling, respectively. SAT was more sensitive to ADX and OVX stimuli than VAT. ADX, but not OVX, exerted great effects on reversing the intrinsic difference between SAT and VAT, and consequently promoted the predominant senescent cell accumulation in SAT, which in turn caused immune cell, especially the B cells, infiltration and consequently induced inflammation, selectively in SAT. Both OVX and ADX enhanced the circadian rhythmicity in VAT, and impaired neurogenesis, angiogenesis, tissue morphogenesis, as well as extracellular matrix development in SAT, thus causing dysfunction of adipose tissues and concomitant metabolic disorders.

## 4. Materials and Methods

### 4.1. Animal

Six-week-old female C57BL/6J mice were purchased from the HuaXi Laboratory Animal Center of Sichuan University. All experiment animals were housed in a humidity-controlled 12:12 h light/dark cycle animal facility at 22 ± 2 °C and had ad libitum access to normal chow food and water. The mice were randomly divided into three groups, with 10 mice in each group. then, the sham procedure (sham control), bilateral ovariectomy (OVX) or adrenalectomy (ADX) were performed on the mice in each group under anesthesia. ADX mice were then supplied with 0.9% saline in drinking water to maintain salt homeostasis. All animal experiments were conducted in accordance with the institutional guidelines for laboratory animals, established by the animal care and use committee of Sichuan Agricultural University (No: 20220106).

### 4.2. Fasting Glucose Measurement

At the age of 29 weeks, all mice were fasted for 18 h, then approximately half a drop of blood was drawn from each mouse’s tail. Blood glucose was measured using a Glucometer Elite (Bayer).

### 4.3. Blood Chemical Analysis

Circulating FSH (CSB-E06871m, CUSABIO, MD, USA), corticosterone (KGE009, R&D Systems, MN, USA) and 17β-estradiol (KGE014, R&D Systems, MN, USA) were determined with commercial ELISA kits, according to the manufacturer’s instructions. Serum free fatty acid (FFA) and triglyceride (TG) levels were measured via a nonesterified free fatty acids assay kit (A042-2-1) and triglyceride assay kit (A110-1-1) from Nanjing Jiancheng Bioengineering Institute (Nanjing, China).

### 4.4. Hematoxylin-Eosin (H&E) Staining

Fresh adipose tissue samples were soak in 10% neutral buffered formalin (G2161 Solarbio, Beijing, China) for 48 h, and then embedded in paraffin. Then, 5-μm-thick sections were sliced. Standard Hematoxylin-Eosin staining was then performed with a H&E stain kit (G1120 Solarbio, Beijing, China). The adipocyte area was then calculated using Image J software (National Institutes of Health, Bethesda, MD, USA), as described previously [17].

### 4.5. Senescence-Associated β-Galactosidase (SA-β-Gal) Staining

SA-β-gal activity was measured in fresh adipose tissue with a senescence β-galactosidase staining kit (C0602, Beyotime, Shanghai, China). Briefly, 50 mg fresh adipose tissue chunks were collected, fixed with fixative solution for 20 min, then washed 3 times in PBS, and incubated in 750 μL staining solution with X-gal for 14 h at 37 °C in a CO_2_—free incubator. The adipose tissue chunks were then washed in 70% EtOH and PBS. The image was captured using a CanoScan LiDE 300 (Canon).

### 4.6. RNA Sequencing

Total RNA was extracted from the crushed adipose tissue samples using TRIzol™ Reagent (Invitrogen, Waltham, MA, USA). RNA quantity and quality were determined by using RNA Nano 6000 Assay Kit of the Bioanalyzer 2100 system (Agilent Technologies, CA, USA).Then, 1000 ng of total RNA per sample with RNA integrity numbers (RINs) greater than or equal to 8.0 were used as the input material to prepare the RNA sample. Briefly, mRNA was purified from total RNA using oligo (dT) magnetic capture beads. The libraries were synthesized by using NEBNext UltraTM RNA Library Prep Kit for Illumina (NEB, Ipswich, MA, USA), according to the manufacturer’s instructions. Then, the cDNA library was sequenced on an Illumina NovaSeq platform and generated 150 bp paired-end reads. Fastp (Version 0.19.7) was used to filter the low quality reads (defined as reads with more than 50% beads scoring Qphred ≤20) and to remove the adapter sequence to generate clean reads. The mouse reference genome (http://ftp.ensembl.org/pub/release-105/fasta/mus_musculus/dna/, accessed on 12 April 2022) and gene annotation files (http://ftp.ensembl.org/pub/release-105/gtf/mus_musculus/, accessed on 12 April 2022) were downloaded from Ensembl directly to build the index of the reference genome, then the clean reads were then aligned to the reference genome using Hisat2 (v2.0.5). For the gene expression level quantification, featureCounts (v1.5.0-p3) was used to count the read numbers mapped to each gene. The R package DESeq2 (1.20.0) was used to perform the pairwise comparisons among the groups, the resulting *p*-values were adjusted using the Benjamini and Hochberg’s approach for controlling the false discovery rate (FDR), the genes that met the criteria of FDR < 0.05 were considered as a differentially expressed genes (DEGs).

### 4.7. Functional Enrichment Analysis

Database for Annotation, Visualization and Integrated Discovery (DAVID) bioinformatics database (2021 update) [19] was used to perform the gene ontology (GO) and the Kyoto Encyclopedia of Genes and Genomes (KEGG) analysis. For all GO terms and KEGG pathways, a threshold of FDR < 0.05 was set for the significance.

### 4.8. Protein-Protein Interaction Network Analysis

Protein-protein interaction network was built up using STRING APP of Cytoscape (3.9.1), the minimum required interaction score was set as 0.4. Cytoscape (3.9.1) was used for the network visualization.

### 4.9. Analyzing the ChIP-Seq Data

The chromatin immunoprecipitation sequencing (ChIP-seq) data of the PPARG-binding sites on murine 3T3L-1 cells (SRX330315) was obtained and the conducted peak calling with a threshold of q < 1 × 10^−5^ by ChIP-Atlas (https://chip-atlas.org, accessed on 20 October 2022) was determined [69]. The R package ChIPseeker [70] was employed for the peak annotation, the promoter region was designated as ±3 kb of the transcription starts site (TSS), as recommended by the authors.

### 4.10. Inferring the Transcription Factor Activity

The transcription factor enrichment assay was performed using ChIP-X Enrichment Analysis Version 3 (ChEA3) (https://maayanlab.cloud/chea3/, accessed on 25 October 2022) [42]. ChEA3 used the ChIP-seq results from ENCODE, ReMap and from the literature, TF-target co-expression data from the ARCHS4 and GTEx databases, and the TFs target co-occurrence in Enrichr queries, and then the mean rank was calculated, and the criteria of the mean rank < 30 was used for the TF selection, as recommended by the authors.

### 4.11. Immune Cell Infiltration Analysis

CIBERSORTx (https://cibersortx.stanford.edu/, accessed on 28 October 2022) [47] was used to measure the relative proportions of the infiltrating immune cells in adipose tissue, based on the gene expression profile, the signature matrix file of the immune cells was built, based on the mouse tissue expression profiles [62]. The number of permutations for the statistical analysis was set as 1000.

### 4.12. Statistical Analysis

All statistical analyses was performed with GraphPad Prism 9 software (La Jolla, CA, USA). All statistical tests are fully described in the figure legends and met the criteria for the normal distribution with a similar variance. Briefly, the two-tailed Student’s t test was conducted to compare two groups. For comparison between more than two groups, a one-way ANOVA followed by Tukey’s test were used. All values were presented as mean ± SEM. Significance is given as * *p* < 0.05, ** *p* < 0.01 and *** *p* < 0.001.

## Figures and Tables

**Figure 1 ijms-24-02308-f001:**
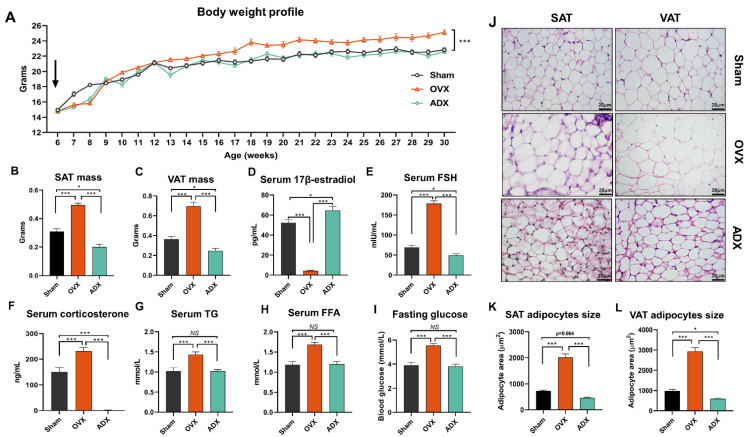
Opposite response in endocrine hormones and the AT expansion caused by ADX and OVX. (**A**) The body weight profile of the sham control, OVX and ADX mice across the entire experimental periods. (**B**,**C**) SAT and VAT weight of the sham control, OVX and ADX mice. (**D**–**I**) Serum 17β-estradiol, FSH, corticosterone, triglyceride (TG), free fat acid (FFA) and fasting blood glucose concentrations of the sham control, OVX and ADX mice. (**J**) Representative histological images of SAT and VAT of the sham control, OVX and ADX mice. Scale bar: 20 μm. (**K**,**L**) Average adipocytes size of SAT and VAT of the sham control, OVX and ADX mice. All data met the criteria for normal distribution with similar variance. Statistical analysis was performed using repeated-measures 2-way ANOVA with Holm–Sidak’s multiple comparisons (**A**) and one-way ANOVA, followed by Tukey’s test (**B**–**E**,**F**–**L**). Data are expressed as mean ± SEM. *NS*, not significant, * *p* < 0.05, *** *p* < 0.001.

**Figure 2 ijms-24-02308-f002:**
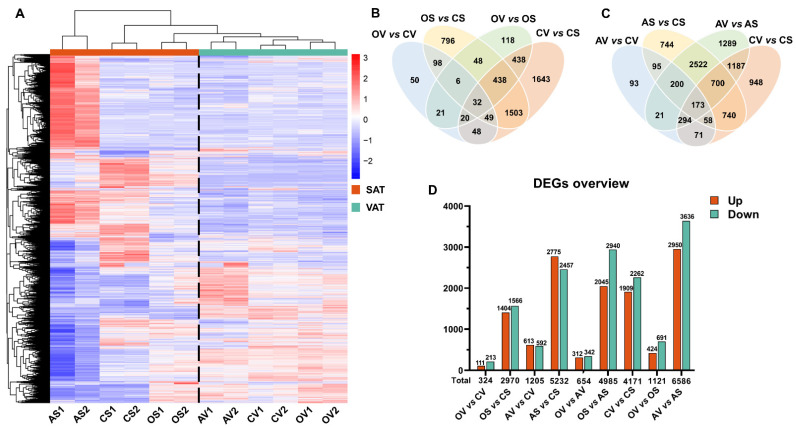
Distinct transcriptomic response of AT to OVX versus ADX in female mice. (**A**) Hierarchical clustering analysis of the differentially expressed genes (DEGs), based on the z-score of their FPKM value. (**B**) Venn diagram of differentially expressed genes between SAT and VAT of OVX and the sham control mice. (**C**) Venn diagram of DEGs between SAT and VAT of ADX and the sham control mice. (**D**) Overview of DEGs among groups. AS: SAT of ADX mice; CS: SAT of the sham control mice; OS: SAT of OVX mice; AV: VAT of ADX mice; CV: VAT of the sham control mice; OV: VAT of OVX mice.

**Figure 3 ijms-24-02308-f003:**
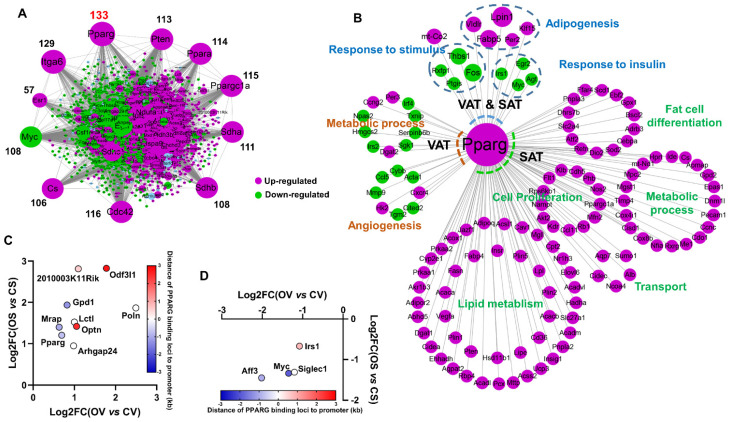
Integrated analysis reveals *Pparg* as a key mediator of the AT expansion in OVX female mice. (**A**) Protein-protein interaction (PPI) analysis of all DEGs induced by OVX in both VAT and SAT. (**B**) Subnetwork of DEGs centered on *Pparg*. Functional enrichment analysis of DEGs was conducted by DAVID (2021 updated). (**C**) OVX upregulated putative *Pparg* target genes in both SAT and VAT. (**D**) OVX downregulated putative *Pparg* target genes in both SAT and VAT.

**Figure 4 ijms-24-02308-f004:**
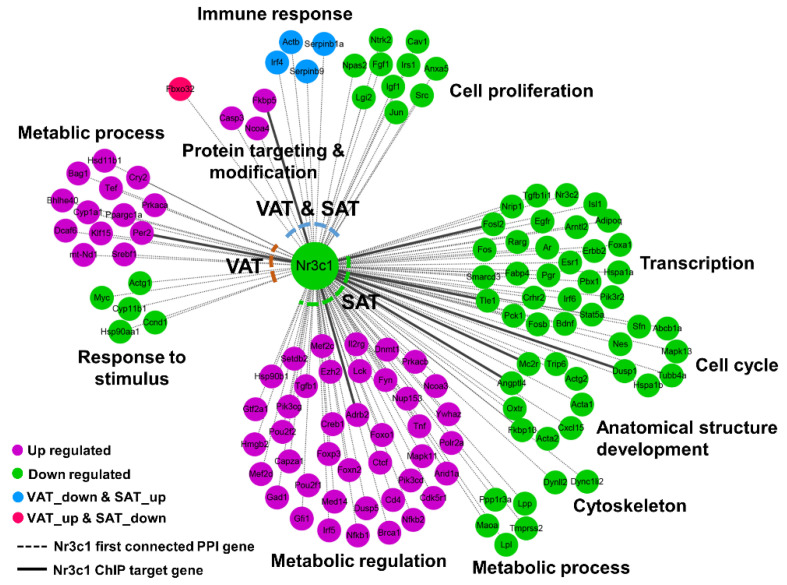
Integrated network analysis of *Nr3c1* first connected the DEGs in both VAT and SAT.

**Figure 5 ijms-24-02308-f005:**
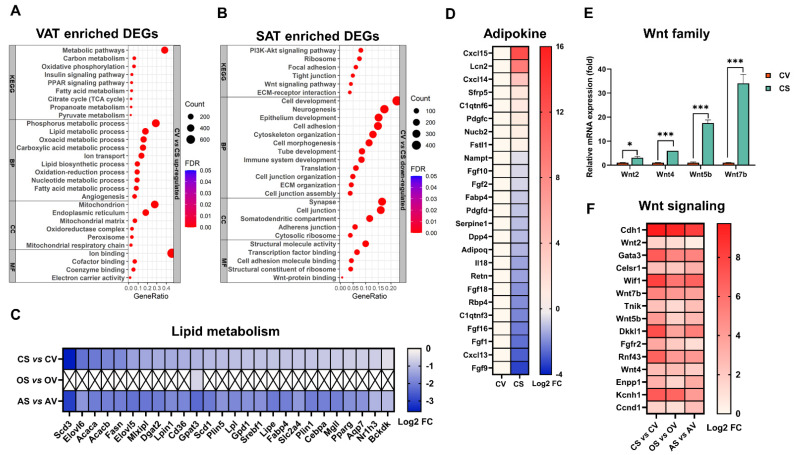
Instinct difference between SAT and VAT in female mice. (**A**) Functional enrichment analysis of the VAT enriched genes (more highly expressed DEGs in VAT than in SAT) in the sham control mice. (**B**) Functional enrichment analysis of the SAT enriched genes (more highly expressed DEGs in SAT than in VAT) in the sham control mice. (**C**) Heatmap of the lipid metabolism associated DEGs between SAT and VAT in the sham control, OVX and ADX mice. (**D**) Heatmap of the adipokine-coding DEGs between SAT and VAT in the sham control mice. (**E**) Expression levels of the Wnt family genes in VAT and SAT of the sham control mice. Data met the criteria for the normal distribution with a similar variance, and a statistical analysis was performed using two-tailed Student’s *t*-test. (**F**) Heatmap of the Wnt signaling associated DEGs between SAT and VAT in the sham control, OVX and ADX mice. Data are expressed as mean ± SEM. *NS*, not significant, * *p* < 0.05, *** *p* < 0.001. KEGG, Kyoto encyclopedia of genes and genomes; BP, biological process; CC, cellular component; MF, molecular function.

**Figure 6 ijms-24-02308-f006:**
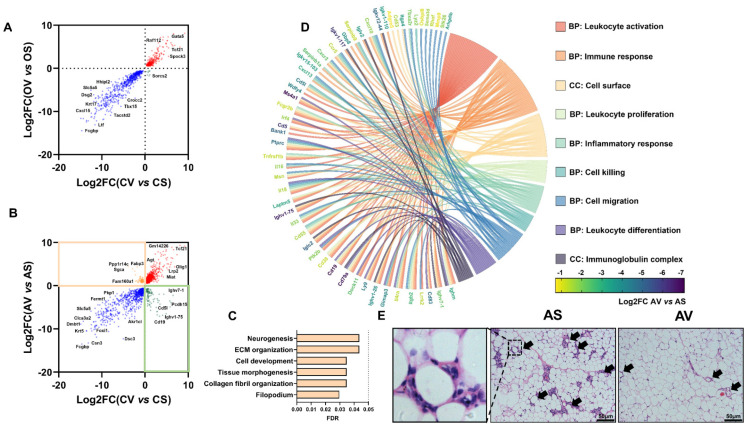
ADX, but not OVX, reverses the intrinsic difference between SAT and VAT in female mice. (**A**) The common regulated genes in SAT and VAT by OVX. (**B**) The common regulated genes in SAT and VAT by ADX. (**C**) Functional enrichment analysis of DEGs which originally were less expressed in VAT than SAT in the sham controls, but reversed by ADX. (**D**) Functional enrichment analysis of DEGs which originally were highly expressed in VAT than SAT in the sham controls, but reversed by ADX. (**E**) Representative histological images of SAT and VAT of ADX mice. Arrows indicate crown-like structures. Scale bar: 50 μm.

**Figure 7 ijms-24-02308-f007:**
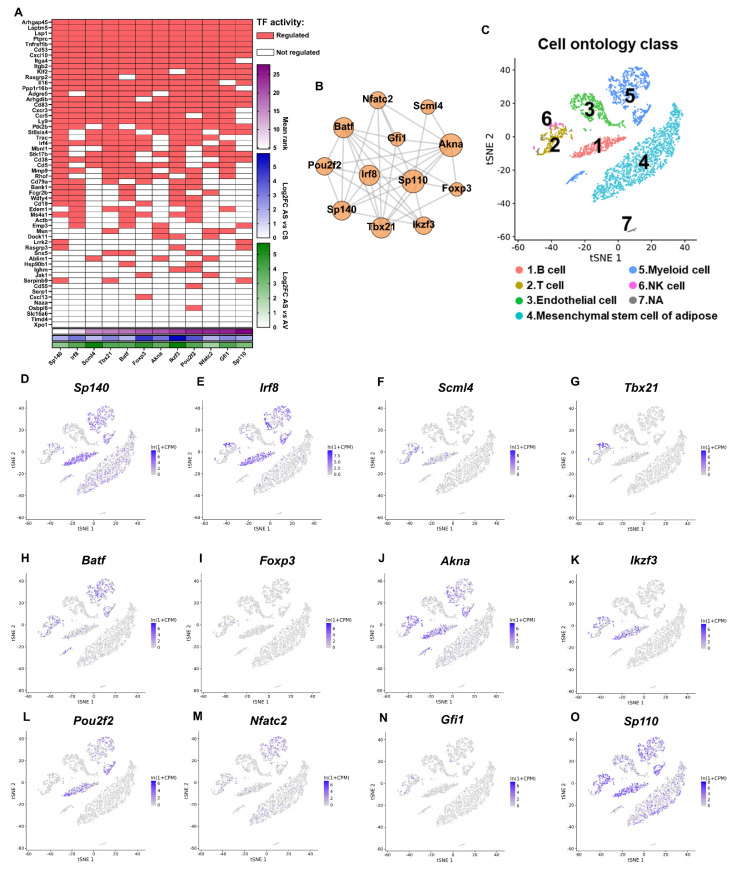
ADX-reversed and inflammation-associated DEGs are mainly transcriptionally regulated by 12 transcription factors. (**A**) Inferred transcriptional regulators of 93 ADX-reversed and inflammation-associated genes via ChEA3. The criteria of the mean rank <30 was used for the TF selection, as recommended by the authors. (**B**) TF-TF co-regulatory networks of the 12 key TFs. The network was constructed by ChEA3 and visualized by Gephi (0.93), and the size of the nodes represents the degree of connectivity of the nodes. (**C**–**O**) Expression levels of *Sp140*, *Irf8*, *Scml4*, *Tbx21*, *Batf*, *Foxp3*, *Akna*, *Ikzf3*, *Pou2f2*, *Nfatc2*, *Gfi1* and *Sp110* in C57BL/6 mouse adipose tissue. The scRNA-seq data was acquired from *Tabula Muris* (https://tabula-muris.ds.czbiohub.org/, accessed on 6 October 2022 ).

**Figure 8 ijms-24-02308-f008:**
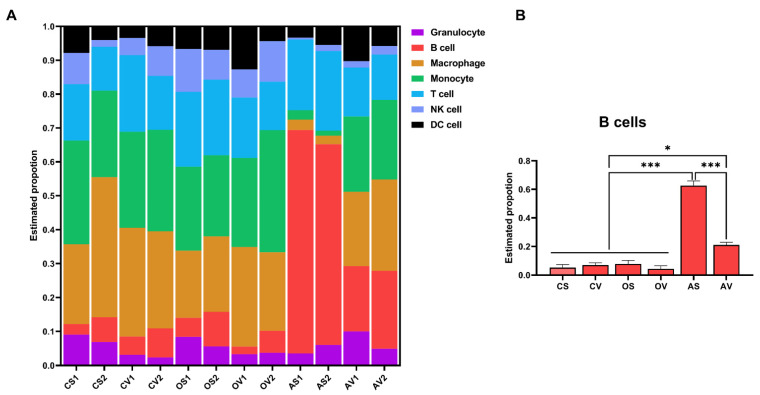
CIBERSORTx algorithm analysis indicates that ADX caused the massive B cell infiltration in SAT. (**A**) The estimated proportion of immune infiltrating cells in the SAT and VAT of the sham control, OVX and ADX mice. (**B**) The estimated proportion of B cells in SAT and VAT of the sham control, OVX and ADX mice; data met the criteria for the normal distribution with a similar variance and a statistical analysis was performed using a one-way ANOVA followed by Tukey’s test. Data are presented as mean ± SEM. NS, not significant, * *p* < 0.05, *** *p* < 0.001.

**Figure 9 ijms-24-02308-f009:**
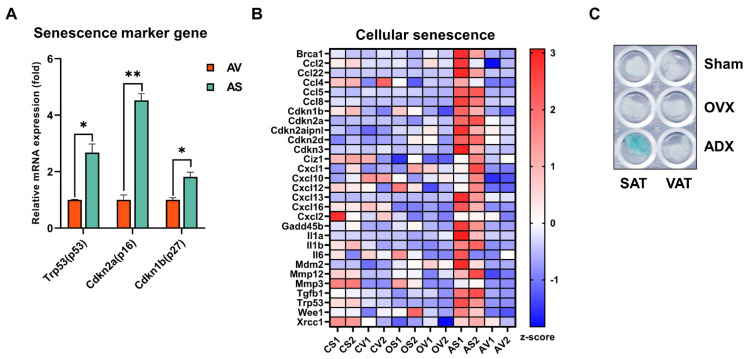
ADX promoted senescent cell accumulation in SAT. (**A**) Expression levels of senescence marker genes in VAT and SAT of ADX mice; data met the criteria for the normal distribution with similar variance and a statistical analysis was performed using a two-tailed Student’s *t*-test. (**B**) Heatmap analysis of the DEGs involved in cellular senescence based on the z-score of their FPKM value. (**C**) Senescence-associated β-galactosidase (SA-β-gal) stain assay of SAT and VAT of the sham control, OVX and ADX mice. Data are presented as mean ± SEM. NS, not significant, * *p* < 0.05, ** *p* < 0.01.

**Figure 10 ijms-24-02308-f010:**
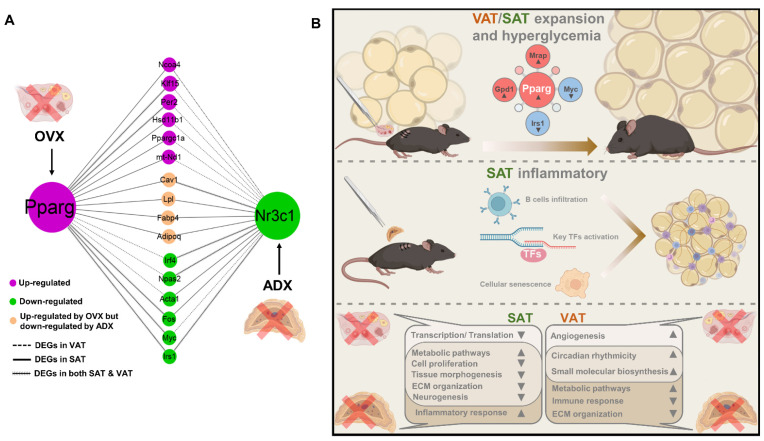
The distinct and common roles of ADX and OVX in regulating the heterogeneous WAT remodeling. (**A**) Crosstalk of the *Pparg* signaling and *Nr3c1* signaling in regulating the AT remolding. (**B**) Summary of the distinct and common roles of OVX versus ADX in regulating the SAT/VAT remodeling.

## Data Availability

All data generated or analyzed during this study are included in this published article and its Appendix A. The RNA sequencing raw data generated during the current study is available in the GSA database (https://bigd.big.ac.cn/gsa/browse; Accession: CRA009293, accessed on 22 December 2022).

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
