# Peer review of "Mechanic Insight into the Distinct and Common Roles of Ovariectomy Versus Adrenalectomy on Adipose Tissue Remodeling in Female Mice"

_ijms, 2023, doi:10.3390/ijms24032308_

Round 1

Reviewer 1 Report

Investigation of the impact of ovarectomy or adrenalectomy, as model situations for the loss of estrogens or corticosteroids, on fat remodeling and distribution is worthwhile with respect to harmful effects of obesity. Though there exist reports addressing this issue, the value of the paper according to the reviewer consists in its comprehensiveness and complexity. I appreciate a complex view on the problematics and a large work.

There are only a few errors  to be corrected:

 l.55: .. Patients with Addison’s disease (hypercortisolism)

l. 78: Heading should  be Results

l. 445, “mianly” should be “mainly”

l.500… Tbale” should  be “Table”

l. 674-675: “ADX  but OVX exerted a profound effect on the intrinsic difference..” should  probably be “but not OVX.”

l. 685… “Func-tional” should be “functional”

l.700: “pivtiol” should be “pivotal”

Author Response

Dear Reviewer, we highly appreciated you taking the time to review our manuscript so patiently and helping us find so many detailed language errors. All these errors you pointed out have been corrected strictly as your good suggestions (marked in red in the revised manuscript). And, we again carefully reviewed and fixed all grammatical problems throughout the manuscript.    

Special thanks to your good comments and suggestions!

Reviewer 2 Report

Title: Mechanic insight into the distinct and common roles of ovariectomy versus adrenalectomy on adipose tissue remodeling in female mice.

Authors: Weihao Chen, Fengyan Meng, Xianyin Zeng, Xiaohan Cao, Guixian Bu, XiaoGang Du, Guozhi Yu, Fanli Kong, Yunkun Li, Tian Gan, and Xingfa Han

General Comment:

Adipose tissue, being an important endocrine organ itself, is , on the other hand, under the impact of other hormones that may modulate its development and function. In their work, Weihao Chen et al. investigated an impact of ovariectomy - and adrenalectomy-induced deficiency of estrogens and cortisol on clinical phenotype and adipose tissue gene expression in mice. They found that estrogen deficiency, via stimulation of PPARγ signaling, promoted adipocyte differentiation and fat accumulation in both subcutaneous and visceral adipose tissue depots, while lack of glucocorticoid signaling inhibited adipocyte proliferation and led to lipoatrophy in both depots. Moreover, adrenalectomy was associated with the accumulation of senescent cells and the development of inflammatory response in subcutaneous tissue only. Both interventions enhanced circadian rhythmicity in visceral depot while impairing cell proliferation, neurogenesis, tissue morphogenesis, and extracellular matrix organization in subcutaneous adipose tissue selectively. The presented work is an example of a complex approach to the research task, combining in vivo, in vitro, and in silico experiments. Therefore, I have only some minor remarks the authors could kindly address before the manuscript is accepted for publication.

Minor revisions:

- Statistics – in figures' legends, please declare if the data distribution in the studied groups was normal – then the use of parametric tests and presenting data as SEMs is justified

- Editing – please use a hyphenation function in the whole Results section

- Line 585 – Please replace "phonotypes" with "phenotypes."

Author Response

General Comment response:

We highly appreciated your positive comments on our research work.

Minor revisions' response:

1) - Statistics – in figures' legends, please declare if the data distribution in the studied groups was normal – then the use of parametric tests and presenting data as SEMs is justified

Response: Done as suggested (marked in red in the revised manuscript).

2) - Editing – please use a hyphenation function in the whole Results section

Response: Have corrected as suggested.

3) - Line 585 – Please replace "phonotypes" with "phenotypes".

Response: Have corrected as suggested (marked in red in the revised manuscript).